# Anti-Bacterial, Anti-Viral, and Anti-Inflammatory Properties of Kumazasa Extract: A Potential Strategy to Regulate Smoldering and Inflammation

**DOI:** 10.3390/medicina61091511

**Published:** 2025-08-22

**Authors:** Hideki Iwasaki, Shirol Gulam, Tomoji Maeda, Mineo Watanabe, Tokuko Takajo, Soh Katsuyama, Hiroaki Sano, Takanari Tominaga, Akio Ogawa, Ken-ichi Sako, Toru Takahashi, Takahiro Kawase, Takamitsu Tsukahara, Yoshikazu Matsuda

**Affiliations:** 1Division of Clinical Pharmacology, Graduate School of Pharmaceutical Science, Nihon Pharmaceutical University, Ina 362-0806, Japan; 217008@ym.nichiyaku.ac.jp (H.I.); 257004@ym.nichiyaku.ac.jp (S.G.); t-maeda@nichiyaku.ac.jp (T.M.); m-watanabe@nichiyaku.ac.jp (M.W.); takajo@nichiyaku.ac.jp (T.T.); soukatsuyama@nichiyaku.ac.jp (S.K.); sakok@nichiyaku.ac.jp (K.-i.S.); t-takahashi@nichiyaku.ac.jp (T.T.); 2Raffinee International Inc., Tokyo 103-0023, Japan; sano@raffinee.co.jp (H.S.); tominaga@raffinee.co.jp (T.T.); ogawa@raffinee.co.jp (A.O.); 3Kyoto Institute of Nutrition and Pathology, Kyoto 610-0231, Japan; kawase@kyoto-inp.co.jp (T.K.); tsukahara@kyoto-inp.co.jp (T.T.)

**Keywords:** Kumazasa extract, anti-bacterial effect, anti-influenza virus effect, anti-inflammatory effect, in vitro, in vivo

## Abstract

*Background and Objectives*: Kumazasa extract (KZExt) is a food product obtained by steam extraction of *Kumazasa* (*Sasa senanensis* and *Sasa kurilensis*) leaves under high temperature and pressure. It contains abundant polyphenols, including *trans*-p-coumaric acid and ferulic acid, as well as xylooligosaccharides. In this study, we investigated the antibacterial, anti-viral, and anti-inflammatory effects of KZExt in vitro and in vivo. *Materials and Methods:* The anti-oxidant, antibacterial, and anti-viral effects of KZExt were assessed in vitro. Anti-oxidant activity was evaluated based on the scavenging of •OH, •O_2_^−^ and ^1^O_2_. Antibacterial effects were assessed by determining the minimum inhibitory concentration (MIC) using a microdilution method. Anti-influenza activity was measured via plaque formation in MDCK cells. Anti-inflammatory effects were assessed by measuring interleukin (IL)-1β inhibition in lipopolysaccharide (LPS)-stimulated RAW264.7 cells. In vivo, KZExt was administered once (30 min before) in a formalin-induced inflammation model to evaluate pain-related behavior. In the LPS-induced inflammation model, KZExt was administered for five days before LPS injection. Behavioral changes and cytokine levels were assessed 24 h later via the open field test and cytokine quantification. *Results*: In vitro, KZExt showed antibacterial, anti-influenza, and anti-oxidant effects, and suppressed LPS-induced IL-1β production. In vivo, it significantly reduced the second phase of formalin-induced pain behavior. In the LPS model, although behavioral changes were unaffected, KZExt suppressed IL-6 and interferon-γ production. *Conclusions*: The antibacterial, anti-viral, and anti-inflammatory effects of KZExt were confirmed in vitro and in vivo. Notably, the anti-inflammatory effect suggests potential immunomodulatory activity. These findings indicate that KZExt may help suppress smoldering inflammation and inflammation associated with various diseases through its combined antibacterial, anti-viral, and immunomodulatory actions.

## 1. Introduction

Kumazasa extract (KZExt) is a food product obtained via steam extraction of *Kumazasa* (*Sasa senanensis* and *Sasa kurilensis*) leaves under high-temperature and high-pressure conditions. When using the mist extraction method, the product contains more polyphenols, such as t-p-coumaric acid and ferulic acid, than when extracted by alkali treatment, ethanol extraction, and hot water extraction [1,2]. In addition, it contains xylooligosaccharides [1,2]. Various functionalities have been reported for bamboo, including Kumazasa, such as UV protection [3], antiviral activity [4], anti-oxidant activity [5], inhibition of amyloid beta-mediated neuronal cell damage [6], and inhibition of macrophage-derived nitric oxide (NO) and prostaglandin E_2_ (PGE_2_) production [7,8]. These reports suggested that KZExt may have anti-inflammatory effects similar to those of bamboo.

Chronic inflammation is recognized as one of the most significant and complex health challenges in modern medicine [8,9]. It is characterized by a low-grade, persistent inflammatory state triggered by continuous exposure to various stimuli and stressors. Over time, this condition contributes to the development and progression of numerous major chronic diseases, including cardiovascular disease (CVD), metabolic syndrome, type 2 diabetes, Alzheimer’s disease, Parkinson’s disease, and cancer [10,11,12]. The underlying mechanisms of chronic inflammation include sustained overactivation of the immune response, excessive production of inflammatory cytokines (e.g., interleukin-1β [IL-1β], interleukin-6 [IL-6], and tumor necrosis factor-α [TNF-α]), overproduction of reactive oxygen species (ROS), and the resulting cellular damage and tissue dysfunction [13,14,15]. Infections are also considered risk factors for chronic inflammation, as they can initiate and perpetuate inflammatory responses. Due to the potential side effects of long-term use, anti-inflammatory drugs are not a practical solution for managing chronic inflammation. As a result, there is growing interest in alternative approaches, such as dietary interventions.

KZExt is already marketed as a food product in Japan, with no reported safety concerns. However, its in vitro and in vivo biological activities have not yet been thoroughly evaluated. Therefore, in this study, we investigated the antibacterial, antiviral, and anti-inflammatory properties of KZExt through both in vitro and in vivo experiments. In particular, its anti-inflammatory effects were assessed using an LPS-induced inflammation model, which we previously reported as a representative model of smoldering inflammation [16].

## 2. Materials and Methods

### 2.1. KZExt

KZExt (Kumazasa Extract Premium, Lot No. 202508) was provided by Raffine International Co., Ltd. (Tokyo, Japan).

### 2.2. In Vitro Evaluations

#### 2.2.1. Evaluation of the Anti-Microbial Effect of KZExt

The minimum inhibitory concentration (MIC) of KZExt against each bacterial strain was determined using a microdilution method. Briefly, 100 μL of a liquid medium suitable for MIC determination of each bacterium or fungus was added to a 96-well flat-bottom plate. A two-fold serial dilution of KZExt was prepared. Table 1 shows the bacterial and fungal strains, media, and culture conditions. Each bacterial strain was suspended in sterile water, and the turbidity was adjusted according to the McFarland No. 0.5 standard. Then, 10 μL of the adjusted suspension was added to each well. Following incubation, bacterial growth was visually assessed, and the MIC was defined as the lowest concentration at which no visible growth was observed.

#### 2.2.2. Evaluation of the Anti-Influenza Virus Effect of KZExt

The anti-influenza virus effects of KZExt were evaluated using a plaque assay. Influenza virus strain JNBP-HUZ001 (PR8) was used in this study. The viral suspension was mixed with KZExt to obtain a final concentration of 10% (*v*/*v*), followed by incubation at 37 °C for 30 min. The suspension was then serially diluted tenfold in Hank’s Balanced Salt Solution (HBSS) containing 20 μg/mL of trypsin (Type III). Confluent monolayers of MDCK cells in a 60-cm^2^ cell culture dish were washed twice with HBSS, followed by the addition of 100 μL aliquots of the serially diluted viral suspension to each dish. After incubation at 37 °C for 30 min, the dishes were washed twice with HBSS, and 2 mL of overlay medium (Eagle’s minimum essential medium supplemented with 2% FCS, 0.45% agarose, 100 U/mL penicillin, and 100 μg/mL streptomycin) was added to each dish. The plates were then incubated at 37 °C for 72 h. After incubation, the soft agar overlay was carefully removed, and 1 mL of 1% crystal violet solution with 10% buffered formaldehyde was added to each dish. After 30 min at room temperature, the dishes were washed with tap water to remove excess stain, and plaques on the blue-purple background were counted.

#### 2.2.3. Evaluation of the Anti-Oxidant Effect of KZExt

The anti-oxidant activity was assessed by measuring the scavenging of hydroxyl radical (•OH), superoxide anion (•O_2_^−^), and singlet oxygen (^1^O_2_), and they were measured with a JES-FA100 ESR spectrometer (JEOL Ltd., Tokyo, Japan). •OH was generated via the Fenton reaction (Fe^2+^ and H_2_O_2_), •O_2_^−^ was produced using the xanthine/xanthine oxidase system, and ^1^O_2_ was generated upon irradiation of a rose bengal solution with visible light. •O_2_^−^ ions were detected via ESR spectroscopy using 5,5-dimethyl-1-pyrroline N-oxide (DMPO, Labotec Co., Ltd., Tokyo, Japan) as a spin-trapping agent. ^1^O_2_ was detected by ESR after reaction with 2,2,5,5-tetramethyl-3-pyrroline-3-carboxamide (TPC, Sigma-Aldrich Co., Ltd., St. Louis, MO, USA).

#### 2.2.4. Evaluation of KZExt on LPS-Stimulated IL-β Production

Lipopolysaccharide (LPS; E. coli O111:B4, Sigma-Aldrich, St. Louis, MO, USA) was used to induce inflammation in RAW 264.7 murine macrophages (Riken, Wako, Saitama, Japan). Cells were seeded at a density of 5 × 10^5^ per well in 24-well plates and incubated at 37 °C with 5% CO_2_ for 24 h. Subsequently, LPS (100 ng/mL) was added with or without KZExt at concentrations of 0.1% or 0.01%, followed by a 48 h incubation. The supernatants were collected, and IL-1β levels were quantified using ELISA kits (Abcam, Cambridge, UK) to assess inflammatory response.

### 2.3. In Vivo Evaluations

#### 2.3.1. Effect of KZExt in a Mouse Model with Formalin-Induced Inflammation

To evaluate the potential anti-inflammatory effects of KZExt, a mild acute inflammation model was employed using formalin-induced pain in mice. Male ddY mice (4 weeks old, 20–26 g; Japan SLC, Hamamatsu, Shizuoka, Japan) were housed under controlled conditions (12 h light/dark cycle, 23 ± 1 °C, 52 ± 2% humidity) with free access to food and water. The study was approved by the Nihon Pharmaceutical University Animal Ethics Committee (AE2023-005).

Mice received oral administration of KZExt (0.2 mL/mouse) or distilled water 30 min prior to injection of 20 µL of 2% formalin into the hind paw. Pain-related behaviors (licking and biting) were observed for 30 min post-injection to assess nociceptive responses [16].

#### 2.3.2. Effect of KZExt in Mice Model with LPS-Induced Inflammation

To simulate systemic inflammation, C57BL/6J male mice (8 weeks old; CLEA Japan, Tokyo, Japan) were acclimated for 7 days and randomly assigned to three groups (saline, LPS, LPS + KZExt; N = 7–8/group). To ensure the objectivity of the results, the study was conducted at the Kyoto Institute of Nutritional Pathology, an independent third-party institution. The research was carried out with the approval of the Animal Ethics Committee of the same institute (Approval date: 7 May 2024; Approval number: 23027NY②). From days 2 to 7, KZExt (0.2 mL/20 g BW) or saline was administered orally [17,18]. On day 7, LPS (0.1 mg/mL) was injected intraperitoneally (0.1 mL/20 g BW). Mice were euthanized on day 8, and blood was collected under anesthesia (medetomidine, midazolam, and butorphanol). Plasma was separated and stored at −80 °C [19,20].

#### 2.3.3. Open Field Test (OFT)

Each mouse was placed in a disinfected open field arena (500 mm × 500 mm × 400 mm) under 100 lx lighting. Locomotor activity was recorded for 10 min per mouse at 10:00 AM.

#### 2.3.4. Quantification of Blood Cytokine Levels

Plasma levels of IL-6, IL-10, IL-12 p70, IFN-γ, MCP-1, and TNF were measured using a cytometric bead array kit (Mouse Inflammation CBA; BD Biosciences, Tokyo, Japan), following the manufacturer’s protocol.

### 2.4. Statistical Analyses

Data were analyzed using one-way ANOVA followed by Dunnett’s or Tukey’s post hoc tests. For formalin model data, normality was confirmed and *t*-tests were applied. In the LPS model, behavioral metrics and cytokine levels were compared using Holm’s and Tukey–Kramer tests, respectively. Smirnoff’s test was used to verify data distribution. A *p*-value < 0.05 was considered statistically significant.

## 3. Results

### 3.1. In Vitro Evaluation

#### 3.1.1. Anti-Microbial Effect of KZExt

Table 2 shows the MIC of KZExt against each bacterial strain. KZExt showed antibacterial activity against the foodborne bacteria *Campylobacter* and *Staphylococcus aureus* and *Streptococcus pyogenes*, which causes pharyngitis, at concentrations of 100-fold dilution (1%) or less. Antibacterial activity was also detected against *Escherichia coli*, *Pseudomonas aeruginosa*, and *Salmonella* (MIC%: 1.6–3.1). The MIC against oral anaerobic bacteria (periodontal disease bacteria), such as *P. gingivalis*, and caries-related *Streptococcus mutans* was 6.3%. The antifungal activity (*Aspergillus* and *Candida*) was relatively low (12.5%).

Table 2 shows the MIC of KZExt against each bacterial strain. KZExt showed antibacterial activity against *Campylobacter* and *Staphylococcus aureus*, *Streptococcus pyogenes*, at concentrations of 100-fold dilution (1%) or less. Antibacterial activity was also detected against *Escherichia coli*, *Pseudomonas aeruginosa*, and *Salmonella enterica* (MIC%: 1.6–3.1). The MIC against Streptococcus mutans was 6.3%. The antifungal activity (*Aspergillus niger* and *Candida albicans*) was relatively low (12.5%).

#### 3.1.2. Anti-Influenza Virus Effect of KZExt

The effect of KZXet on plaque formation by the influenza virus is shown in Figure 1. KZExt significantly inhibited plaque formation at 10% and 100% concentrations.

#### 3.1.3. Anti-Oxidant Effect of KZExt

Figure 2 shows the effect of KZExt on hydroxyl radical (•OH), superoxide anion (•O_2_^−^), and singlet oxygen (^1^O_2_). For all reactive oxygen species, significant scavenging effects were observed at 0.025% or more of KZExp.

#### 3.1.4. Effect of KZExt on LPS-Stimulated IL-β Production

The effect of KZExt on IL-1β production when Raw 264.7 cells were stimulated with LPS is shown in Figure 3. KZExt significantly suppressed IL-1β production at concentrations of 0.1 and 0.01%.

### 3.2. In Vivo Evaluation

#### 3.2.1. Effect of KZExt in Mice with Formalin-Induced Inflammation

The effect of KZExt on pain-related behaviors in the formalin-induced inflammation model is shown in Figure 4. KZExt had no effect during the first phase for up to 10 min after formalin injection. However, a significant inhibition of pain behavior was observed in the second phase, from 10 to 30 min after injection.

#### 3.2.2. Effect of KZExt in Mice with LPS-Induced Inflammation

Figure 5 shows the exploratory behavior and the total distance traveled 24 h after LPS administration. LPS administration significantly reduced exploratory behavior and the total distance traveled. KZExt did not affect the LPS-induced reduction in exploratory behavior or the total distance traveled by LPS administration. KZExt had no effect on the duration of time spent at the center.

Figure 6 shows the amounts of IL-6, IL-10, IL-12, IFN-γ, MCP-1, and TNF recorded 24 h after LPS administration. KZExt significantly reduced the levels of IL-6 and IFN-γ that were increased by LPS administration. It also showed a tendency to reduce the amount of TNF.

## 4. Discussion

In this study, the antibacterial, anti-influenza, and anti-oxidant activities of KZExt, along with its inhibitory effect on IL-1β production in LPS-stimulated cells, were confirmed through in vitro evaluation. Additionally, in vivo experiments demonstrated that KZExt suppressed pain-related behaviors in mice with formalin-induced inflammation and reduced cytokine production in mice with LPS-induced inflammation. These findings indicate that KZExt exhibits anti-inflammatory, antibacterial, and antiviral properties.

This study demonstrated that KZExt possesses broad-spectrum antibacterial activity, although the effective concentrations varied depending on the target organism. It also exhibited a mild antifungal effect. The reported mechanisms of action include the following: (1) disruption of the cell membrane, leading to leakage of intracellular contents; (2) inhibition of protein synthesis; and (3) interference with metabolic pathways through enzyme inhibition [21,22].

These findings suggest that topical applications of KZExt—such as skin application or gargling—may exert antibacterial and antiviral effects, depending on the specific bacterial and viral species involved.

The anti-oxidant activity of KZExt was demonstrated by its ability to scavenge various reactive oxygen species (ROS). It effectively neutralized hydroxyl radicals (•OH), which are implicated in cancer, lifestyle-related diseases, and aging, as well as superoxide anions (•O_2_^−^), which play a role in host defense but can contribute to tissue damage during ischemia–reperfusion injury. Additionally, KZExt suppressed the production of singlet oxygen (^1^O_2_), a highly reactive species associated with porphyria and photosensitivity. Furthermore, KZExt inhibited lipopolysaccharide (LPS)-induced interleukin-1β (IL-1β) release in RAW264.7 mouse macrophage cells in a concentration-dependent manner. These results indicate that KZExt possesses anti-inflammatory activity in vitro.

Sato et al. reported that co-culture of LPS and Sasa senanensis leaf extract (KLE) with macrophages expressing both TLR2 and TLR4 significantly suppressed TNF-α and IL-1β production and decreased p-JNK and p-ERK protein expression [23]. This was likely due to components in KLE that inhibit the actions of TLR2 and TLR4, as well as JNK and ERK signaling. Based on this, they concluded that KLE stimulates TLR2/4-mediated cytokine induction while also suppressing LPS-induced inflammatory cytokine production, thereby regulating JNK- and ERK-mediated immune responses. A similar mechanism may underlie the effects of KZExt observed in the present study. Thus, KZExt may exert immunomodulatory effects rather than functioning solely as an anti-inflammatory agent.

Based on these results, KZExt exhibits antibacterial, antiviral, anti-inflammatory, and anti-oxidant activities in vitro, suggesting its potential for preventing and improving chronic inflammation. Therefore, we conducted in vivo evaluations using the same tissues as KZExt. In a formalin-induced inflammation model, KZExt suppressed delayed pain responses. In an LPS-induced systemic inflammation model, KZExt did not significantly affect behavioral changes but significantly suppressed the LPS-induced increase in IFN-γ and IL-6 levels. These results confirmed the anti-inflammatory effects of KZExt in vivo. The in vivo inflammation model used in this study is relatively mild and reversible, and thus may be considered representative of chronic inflammation [17,18]. Although pharmacological treatments for acute and chronic inflammation are often similar, the long-term use of anti-inflammatory drugs in chronic conditions poses safety concerns. Therefore, the use of food-based interventions such as KZExt may offer a meaningful and safer alternative for managing chronic inflammation.

Honda et al. reported that biphasic pain behavior is observed in the formalin-induced inflammation model. The first phase results from the direct activation of sensory nerve endings by formalin, whereas the second phase is associated with inflammation caused by tissue injury, leading to the release of inflammatory mediators, such as prostaglandins, from sensory neurons [24]. KZExt suppressed the second phase of pain behavior, suggesting that it exerts anti-inflammatory effects in vivo. Furthermore, cyclooxygenase (COX) metabolites, IL-1β, and TNF-α are known to play key roles in the formalin-induced inflammation model. In the present study, KZExt inhibited LPS-induced IL-1β production in vitro, suggesting that its effects in the formalin model may be mediated, at least in part, by suppression of IL-1β production. Han et al. further reported that IL-33 and its receptor ST2 are involved in mediating formalin-induced inflammatory pain [25]. Given that IL-33 is a member of the IL-1 cytokine family and shares high sequence homology with IL-1β and IL-18, it remains an intriguing question whether KZExt also modulates IL-33 expression or function. In the LPS-induced inflammation model, KZExt significantly reduced elevated levels of IFN-γ and IL-6 at 24 h post-administration. Although the effects were not statistically significant, KZExt also showed a trend toward suppressing LPS-induced increases in MCP-1 and TNF levels. However, KZExt did not reverse the LPS-induced reductions in locomotor activity or exploratory behavior, suggesting that it may lack central nervous system effects, or that the threshold for behavioral improvement in the open field test (OFT) differs from that required for cytokine suppression. Nevertheless, it is anticipated that continued suppression of inflammatory cytokines may eventually lead to improvements in behavioral outcomes.

Oizumi et al. reported that alkaline extract of the leaves of *Sasa senanensis* Rehder (SE) has antibacterial, anti-viral and anti-inflammatory effects [26,27]. Although the details of the mechanism of antibacterial and antiviral effects are unclear, their investigations suggest that it is highly likely that the mechanism is a direct effect of SE, separate from toxicity [26]. On the other hand, the anti-inflammatory effect of SE has been investigated by multi-omics analysis using metabolomics and DNA arrays [27]. As a result, it was confirmed that human gingival fibroblasts (HGF) were stimulated with IL-1β, and that the levels of amino acids, total glutathione, methionine sulfoxide, and 5-oxoproline decreased, while the DNA expression of AKT, CASP3, and CXCL3 increased, and that this change was restored by co-administration of SE. Therefore, it is possible that the anti-inflammatory effect of SE is mediated by various metabolic pathways involved in cell survival, apoptosis, and leukocyte recruitment [27]. Based on the above findings, Oizumi et al. consider the usefulness of using SE in the dental field. The effects of KZExt observed in this study were similar to those reported by Oizumi et al. Therefore, we believe that similar results will be obtained from multi-omics analysis. On the other hand, most of the results in Oizumi et al.’s report were direct action or in vitro evaluation results. In contrast, this study confirmed the effects of KZExt through in vivo evaluation after oral administration and can be said to be the first paper reporting oral activity. Oizumi et al. reported that an alkaline extract (SE) derived from the leaves of Sasa senanensis (Rehder) possesses antibacterial, antiviral, and anti-inflammatory properties [27]. Its anti-inflammatory effects were further investigated through multi-omics approaches, including metabolomics and DNA microarray analyses [27]. The results showed that stimulation of human gingival fibroblasts (HGF) with IL-1β led to decreased levels of amino acids, total glutathione, methionine sulfoxide, and 5-oxoproline, along with increased expression of genes such as AKT, CASP3, and CXCL3. These alterations were reversed by co-administration of SE. These findings suggest that the anti-inflammatory effects of SE may be mediated through multiple metabolic pathways related to cell survival, apoptosis, and leukocyte recruitment [27]. Based on this evidence, Oizumi et al. concluded that SE exerts its effects via these interconnected biological processes. The effects of KZExt observed in the present study were comparable to those reported for SE by Oizumi et al., suggesting that similar outcomes may be observed in future multi-omics analyses of KZExt.

Meanwhile, Oizumi et al. have proposed potential applications of SE in the dental field, based on its direct local effects. They have also reported that SE alleviates fatigue, possibly through a direct stomachic action. Therefore, the in vitro confirmation of KZExt’s effects—extracted via steam distillation—and the demonstration of its efficacy following oral administration in a reversible inflammation model represent novel findings not previously reported. These findings suggest that KZExt may have clinical potential as a functional food capable of preventing smoldering and chronic inflammation, which are implicated in the pathogenesis of various diseases.

## 5. Conclusions

In conclusion, this study suggests that KZExt possesses antibacterial, antiviral, and immunomodulatory properties, and may help alleviate smoldering inflammation. To validate these findings, future research should investigate its effects in humans, including assessments of clinical efficacy. KZExt contains polyphenolic compounds such as xylooligosaccharides, trans-p-coumaric acid, and ferulic acid, whose individual contributions to the observed effects warrant further investigation. Understanding the roles of these components may provide valuable insights for future therapeutic development.

This study has several limitations. First, as the entire extract was used, further investigation is needed to identify the specific active constituents. Clarifying this will help determine whether the observed effects are attributable to individual components, additive interactions, or synergistic mechanisms. Second, there is a need to establish a more clinically relevant model of chronic inflammation. In this study, formalin- and LPS-induced models were used to simulate chronic inflammatory conditions. Future studies should employ repeated-dose models, such as multiple LPS administrations, to more accurately reflect chronic inflammation and to further evaluate the effects of KZExt. Such models may help elucidate more specific effects of KZExt. Based on these findings, we aim to propose novel strategies for the prevention and management of chronic inflammation.

## Figures and Tables

**Figure 1 medicina-61-01511-f001:**
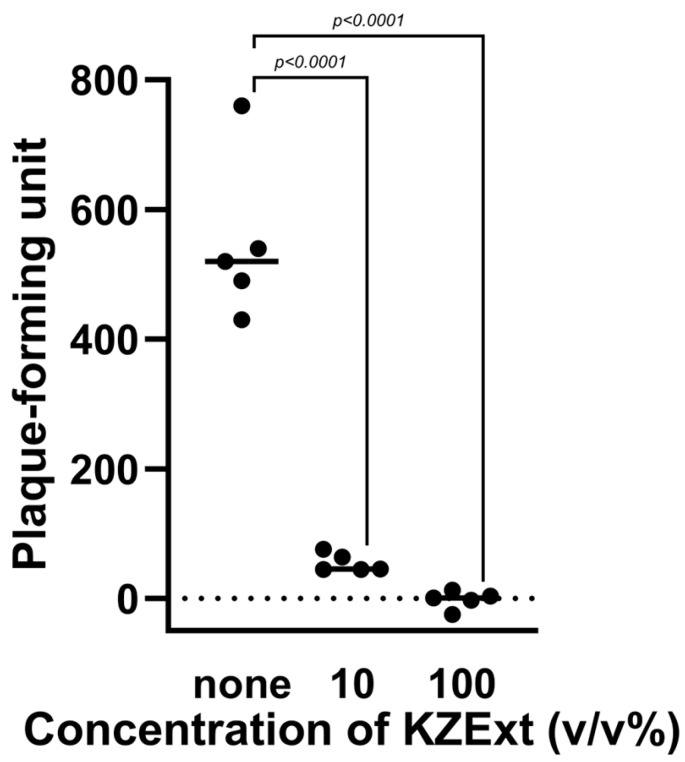
The effect of KZEet on plaque formation by influenza virus. All data are shown as mean ± deviation (S.D.). N = 5.

**Figure 2 medicina-61-01511-f002:**
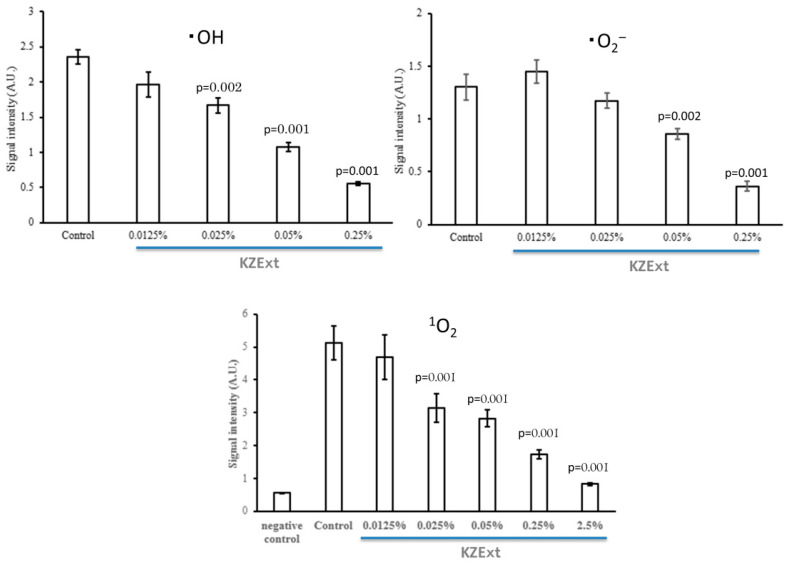
The effect of KZExt on hydroxyl radical (•OH), superoxide anion (•O_2_^−^), and singlet oxygen (^1^O_2_). All data are shown as mean ± error (S.E.). *p* values are reported in comparison with the control group. N = 3.

**Figure 3 medicina-61-01511-f003:**
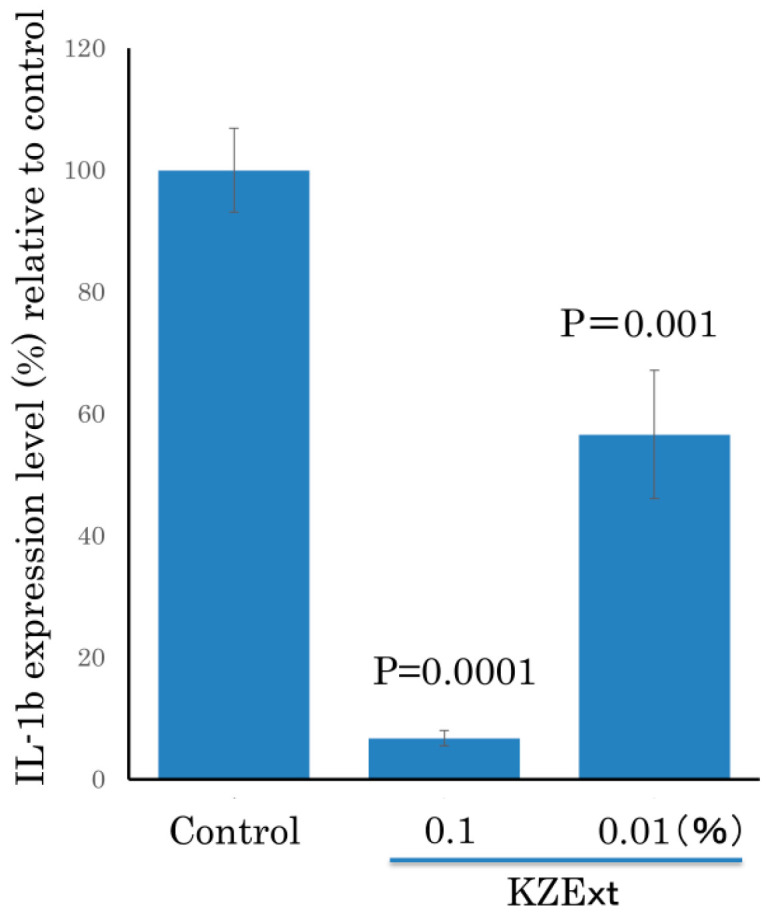
The effect of KZExt on IL-1β production when Raw 264.7 cells were stimulated with LPS. All data are shown as mean ± error (S.E.). *p* values are reported in comparison with the control group. N = 3.

**Figure 4 medicina-61-01511-f004:**
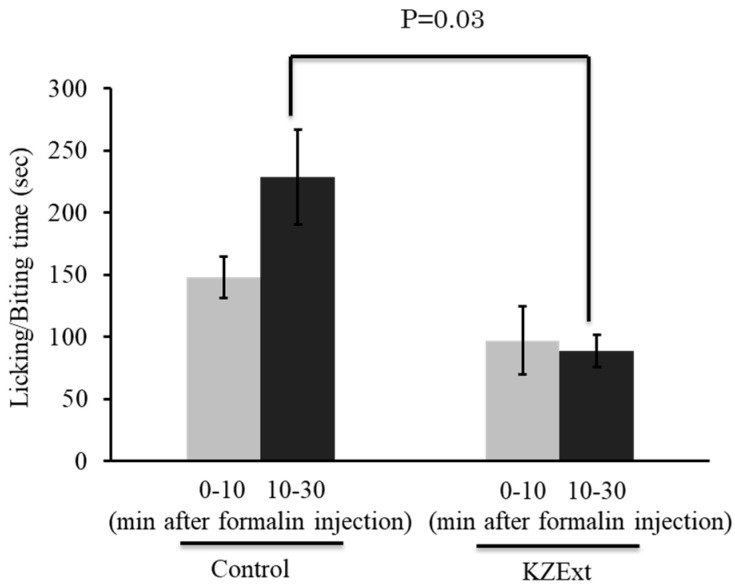
The effect of KZExt on pain-related behavior in mice with formalin-induced inflammation. Comparison between control and KZExt-treated groups at 0−10 and 10−30 min post-injection. *p* values are reported in comparison with the control group. N = 8.

**Figure 5 medicina-61-01511-f005:**
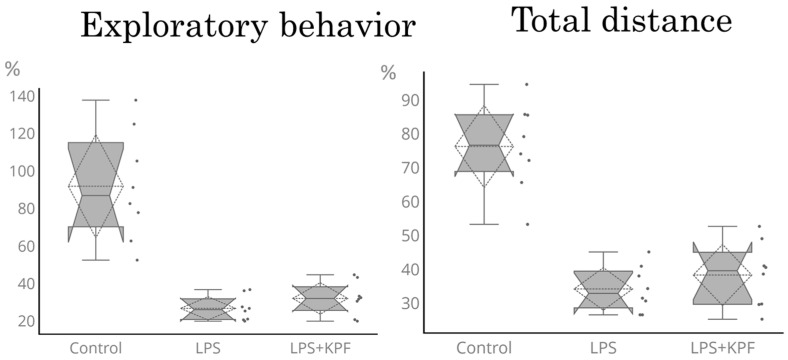
The effect of KZExt on exploratory behavior and total distance traveled 24 h after LPS administration. The data are shown as the relative change in each individual compared to the value before LPS administration. All data are shown as mean ± error (S.E.). N = 8.

**Figure 6 medicina-61-01511-f006:**
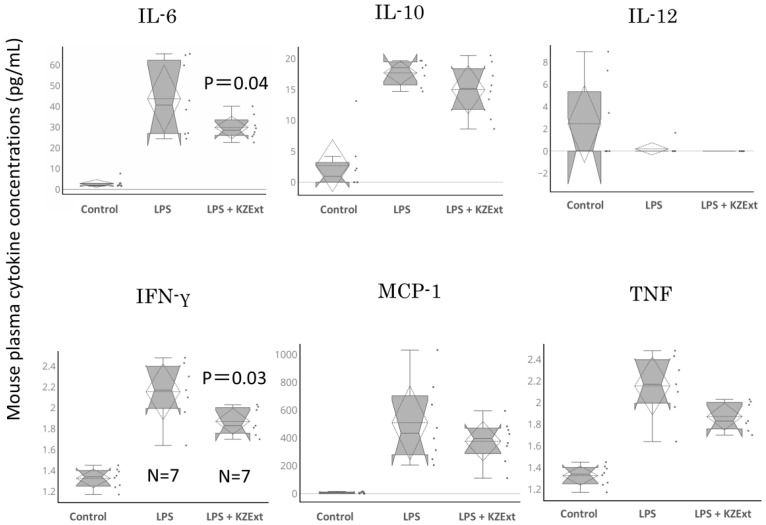
The effect of KZExt on plasma IL-6, IL-10, IL-12, IFN-γ, MCP-1, and TNF levels 24 h after LPS administration. The data are shown as the relative change in each individual compared to the value before LPS administration. All data are shown as mean ± error (S.E.). *p*-values are reported in comparison with the LPS group. N = 7–8.

**Table 1 medicina-61-01511-t001:** Bacterial and fungal strains, media, and culture conditions.

Species	Strain	Culture Media	Culture Condition
*Campylobacter jejuni*	GTC03286	Brain heart infusion broth	Microaerobic, 37 °C, 2 days
*Escherichia coli*	MG1655	Muller Hinton broth with Ca and Mg	Aerobic, 37 °C, 2 days
*Porphyromonas gingivalis*	MW001	Brucella broth with horse blood	Anaerobic, 37 °C, 6 days
*Pseudomonas aeruginosa*	NBRC13275	Muller Hinton broth with Ca and Mg	Aerobic, 37 °C, 2 days
*Salmonella enterica*	NBRC13245T	Muller Hinton broth with Ca and Mg	Aerobic, 37 °C, 2 days
*Staphylococcus aureus*	NBRC13276	Muller Hinton broth with Ca and Mg	Aerobic, 37 °C, 2 days
*Streptococcus mutans*	NBRC19355T	Brain heart infusion broth	Anaerobic, 37 °C, 2 days
*Streptococcus pyogenes*	KI	Muller Hinton broth with Ca and Mg	Aerobic, 37 °C, 2 days
*Aspergillus niger*	NBRC33023	Potato dextrose broth	Aerobic, 30 °C, 3 days
*Candida albicans*	NBRC1385T	Potato dextrose broth	Aerobic, 30 °C, 2 days

Muller Hinton broth was supplemented with Ca^2+^ (25 mg/L) and MG^2+^ (25 mg/L). Brucella broth was supplemented with 5% horse blood.

**Table 2 medicina-61-01511-t002:** Antibacterial activity of KZExt.

Microbes	Minimum Inhibitory Concentration % (*V*/*V*)
*Aspergillus niger*	12.5
*Candida albicans*	12.5
*Porphylomonas gingivalis*	6.3
*Streptococcus mutans*	6.3
*Salmonella enterica*	3.1
*Pseudomonas aeruginosa*	1.6
*Escherichia coli*	1.6
*Streptococcus pyogenes*	0.78
*Staphylococcus aureus*	0.2
*Campylobacter jejuni*	0.1

The bacterial and fungal strains, media, and culture conditions are listed in Table 1.

## Data Availability

The datasets generated and/or analyzed in the current study are available from the corresponding author upon reasonable request.

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
