# Peer review of "Anti-Bacterial, Anti-Viral, and Anti-Inflammatory Properties of Kumazasa Extract: A Potential Strategy to Regulate Smoldering and Inflammation"

_medicina, 2025, doi:10.3390/medicina61091511_

Round 1

Reviewer 1 Report

Comments and Suggestions for Authors

The manuscript titled "Anti-bacterial, Anti-viral, and Anti-inflammatory Properties of Kumazasa Extract: A Potential Strategy to Regulate Smoldering and Inflammation" demonstrates some effort in consolidating the potential effects of Kumazasa extract (KZExt) in vitro and in vivo. However, the study lacks the level of rigor, novelty, and clarity necessary for publication in its current form. 

The primary concern lies in the lack of novelty presented in this manuscript. While the antibacterial, antiviral, and anti-inflammatory effects of Sasa senanensis extracts have been previously reported, including by Oizumi et al. and Sakagami et al. [[36]–[39]], the current study does not sufficiently demonstrate how it advances the field. The claim that this study is the first to evaluate oral activity in vivo is overstated, as similar in vivo results have already been reported for related extracts. Notably, the manuscript fails to explicitly differentiate its findings from existing literature or to identify unique contributions beyond what has already been established.

Another critical issue is the lack of mechanistic insight into the observed effects of KZExt. While the study confirms the antibacterial, antiviral, and anti-inflammatory activities, it provides no data on the molecular pathways or specific bioactive components responsible for these effects. For instance, the antibacterial and antiviral effects are reported but not explored mechanistically, leaving the findings descriptive rather than explanatory. Similarly, the anti-inflammatory effects, particularly the suppression of cytokines such as IL-6 and IFN-γ, are not supported by any mechanistic evaluation, such as pathway analyses or transcriptomic data. Without such insight, the study fails to provide substantial scientific advancement or translational value.

The study design and limitations further undermine the manuscript's conclusions. The in vivo experiments rely on acute inflammation models (formalin-induced and LPS-induced inflammation), yet the study aims to address chronic smoldering inflammation. Acute models are insufficient to draw conclusions about the effects of KZExt on chronic inflammation, and the authors fail to justify the appropriateness of their experimental approach. Moreover, the use of a whole extract rather than isolating and identifying active compounds makes it difficult to attribute observed effects to specific components, hindering reproducibility and translational potential. This is particularly problematic given that the manuscript emphasizes the potential clinical application of KZExt.

Additionally, the data presented are inconsistent and inconclusive. For example, while KZExt reduced cytokine levels (e.g., IL-6 and IFN-γ) in the LPS-induced inflammation model, it failed to reverse behavioral changes such as reduced locomotor activity and exploratory behavior. This inconsistency raises questions about whether the observed anti-inflammatory effects are systemically relevant or limited to isolated biochemical changes. Similarly, the manuscript reports weak antifungal activity, but this result is not contextualized or discussed, leaving its significance unclear.

The manuscript also overemphasizes statistical significance without addressing biological or clinical relevance. While the MIC values against certain bacteria (e.g., Campylobacter jejuni and Staphylococcus aureus) are promising, the study does not discuss whether these concentrations are achievable in practical applications. Similarly, the antioxidant effects of KZExt, although statistically significant, are not compared to standard antioxidants to establish their relative efficacy or relevance under physiological conditions.

The potential for bias in this study is another concern. The research was funded by Raffinee International Inc., the supplier of KZExt, and several authors are employees of this company. While this conflict of interest is disclosed, the absence of independent verification of the results raises questions about the reliability of the findings. Moreover, the manuscript does not address this issue by including validation experiments conducted by independent third-party laboratories.

Finally, the presentation of the manuscript is suboptimal. Figures and tables lack sufficient detail, and the discussion often repeats information already presented in the results section, leading to redundancy. Furthermore, overgeneralized claims, such as the suggestion that KZExt can "ameliorate chronic inflammation" or "contribute to drug discovery," are not substantiated by the data.

In conclusion, the manuscript suffers from significant deficiencies in novelty, experimental design, mechanistic insight, and data interpretation. These issues, combined with concerns about bias and inadequate contextualization of findings, make it unsuitable for publication in its current form. While the study has potential, particularly in its exploration of KZExt's effects on inflammation, substantial revisions, including additional experiments, improved contextualization, and a more balanced discussion, are necessary to bring the manuscript to an acceptable standard.

Reviewer 2 Report

Comments and Suggestions for Authors

This work is devoted to the study of the biological effects of Kumazasa extract (KZExt) obtained from the leaves of Sasa senanensis and Sasa kurilensis plants. The main focus was on the antibacterial, antiviral, and anti-inflammatory activity of the extract, which was studied both in vitro and in vivo. The authors conducted a set of experiments: they evaluated the antioxidant activity through the ability of the extract to neutralize free radicals, determined the minimum inhibitory concentration against bacteria, and also measured the effectiveness of suppressing the influenza virus in MDCK cell culture. The anti-inflammatory effect was evaluated using a model of LPS-induced inflammation in RAW264.7 cells with an emphasis on IL-1b inhibition. An experimental animal model demonstrated a decrease in pain in response to the administration of formalin and a decrease in the levels of pro-inflammatory cytokines IL-6 and interferon-γ with prolonged exposure to KZExt. The integration of in vitro results with data from animal experiments is particularly valuable, which strengthens the validity of conclusions about the anti-inflammatory and immunomodulatory effects of Kumazasa extract. The noted reduction in inflammation and the effect on the immune response may be of practical importance for the development of new natural remedies for the prevention and treatment of inflammatory diseases. In general, the article is executed at a high level, with a clear methodology and a logical presentation of the results. However, for a more complete understanding of the potential of KZExt, it would be worthwhile to additionally consider its pharmacokinetics and safety with prolonged use. The presented work uses 39 sources, of which more than half (20) are over 10 years old. At the same time, a significant part of the publications (sources â„– 10, 13, 14, 15, 18, 19, 20, 22, 23, 24, 25, 26) It is devoted to the issues of aging, obesity, insulin resistance and pathologies of the cardiovascular system. This discrepancy between the subject matter of the sources used and the main focus of the study the study of the antibacterial, antiviral and anti—inflammatory activity of Kumazasa extract - raises questions. Modern scientific practice requires relying on relevant sources that are most relevant to the research topic. The use of a large number of outdated materials and literary sources devoted to rather distant topics reduces the relevance of the literature review and may affect the perception of the novelty and validity of the work. To improve the quality of scientific substantiation, it is recommended to pay more attention to recent studies directly related to the studied effects of Kumazasa extract or related topics, such as current data on the antimicrobial and anti-inflammatory activity of herbal preparations. It is also useful to reduce the number of citations that do not reflect the content of the current study in order to avoid dispersion in scientific reasoning.

Round 2

Reviewer 1 Report

Comments and Suggestions for Authors

Dear author,

After careful revision, the manuscript was revised successfully and can proceed to publication.